# Mycotoxin Biodegradation by *Bacillus* Bacteria—A Review

**DOI:** 10.3390/toxins16110478

**Published:** 2024-11-04

**Authors:** Thanh Nguyen, Xiaojing Chen, Linlin Ma, Yunjiang Feng

**Affiliations:** 1Institute for Biomedicine and Glycomics, Griffith University, Nathan, Brisbane, QLD 4111, Australia; thanh.nguyen4@griffithuni.edu.au (T.N.); linlin.ma@griffith.edu.au (L.M.); 2Bioproton Pty Ltd., Acacia Ridge, Brisbane, QLD 4110, Australia; wendy@bioproton.com; 3School of Environment and Science, Griffith University, Nathan, Brisbane, QLD 4111, Australia

**Keywords:** mycotoxins, contamination, degradation, *Bacillus*, enzyme

## Abstract

Mycotoxins are toxic secondary metabolites produced by various types of fungi that are known to contaminate various food products; their presence in the food chain poses significant risks to human and animal health and leads to enormous economic losses in the food and feed industry worldwide. Ensuring food safety and quality by detoxifying mycotoxin is therefore of paramount importance. Several procedures to control fungal toxins have been extensively investigated, such as preventive measures, physical and chemical methods, and biological strategies. In recent years, microbial degradation of mycotoxins has attracted much attention due to its reliability, efficiency, and cost-effectiveness. Notably, bacterial species from the *Bacillus* genus have emerged as promising candidates for mycotoxin decontamination owing to their diverse metabolic capabilities and resilience in harsh environmental conditions. This review manuscript aims to provide a summary of recent studies on the biodegradation of fungal toxins by *Bacillus* bacteria, thereby illustrating their potential applications in the development of mycotoxin-degrading products.

## 1. Introduction

Mycotoxins are naturally occurring secondary metabolites produced by certain mycotoxigenic fungi [1,2,3]. These natural products are highly toxic and ubiquitously contaminate a number of food categories, including, but not limited to, fruits, vegetables, grains, nuts, seeds, spices, and animal feed, as well as other products, such as herbs, wine, eggs, meat, and milk [3,4,5,6]. The contamination occurs at any stage of the food chain, including pre-harvest, post-harvest, processing, packaging, distribution, and storage of foodstuffs [7]. The presence of mycotoxins in food is regarded as an inherent and unpredictable issue, persisting even with the implementation of good agricultural management and processing practices [8]. This therefore presents a difficult challenge to food safety globally. According to the Food and Agriculture Organization of the United Nations (FAO), it is estimated that approximately 25% of global food products are affected by mycotoxins every year, yet some recent studies have proposed a much higher exposure rate, ranging from 60 to 80% [2,9]; this high level of prevalence contributes to a substantial amount of annual food loss of about one billion metric tons and an outstanding financial loss of billions of dollars in agricultural commodities [5,10]. Not only do mycotoxins challenge agriculture, but they also threaten general public health and wellbeing due to their harmful biological effects, which can cause acute or chronic toxicity in humans and animals [11,12]. Humans can be exposed to mycotoxins through the ingestion of contaminated plant-based foods, the carry-over of mycotoxins into animal-derived products, like meat and eggs, as well as inhalation of air and dust containing these toxins [13]. It is estimated that approximately 3.2 million cases of mycotoxicosis and 50,000 hospitalizations occur annually due to mycotoxins in Europe [14], while aflatoxins alone are responsible for hundreds of hepatocellular carcinoma cases in developing countries each year [8,15]. In animals, mycotoxins primarily enter the body through feedstuffs and forage; they often impair organ functions, such as the liver, kidney, and the immune and reproductive systems, triggering a series of disorders including poor growth, loss of appetite, weight loss, reduced productivity and quality, increased susceptibility to infectious diseases, and sterility [16,17].

In response to the overarching issues posed by mycotoxins, considerable efforts have been undertaken to monitor and mitigate their impacts, ranging from applying traditional preventive measures to employing modern interventive strategies. Various preventive procedures and physical as well as chemical techniques for mycotoxin control have been investigated, such as employing fungicides, using adsorbents, and chemical treatments [2]. Nonetheless, while these methods are able to achieve desired results in eliminating mycotoxins, they often compromise the overall quality of the food or lead to other associated problems [18,19]. Over the past few decades, research interest has increasingly focused on exploring innovative biological approaches to addressing mycotoxin contamination; such emerging methods not only act as alternatives to the conventional strategies because of their low cost, efficacy, and environmental friendliness but also help preserve food shelf life by avoiding interferences and residues from physical and chemical controls [18,20]. Indeed, numerous microorganisms have demonstrated the ability to eliminate and metabolize fungal toxins. For instance, certain lactic acid bacteria are able to inhibit fungal growth and consume aflatoxins, deoxynivalenol, and zearalenone [21,22,23]; some yeast strains are able to degrade patulin, fumonisins, and trichothecenes [24,25]. Notably, accumulating evidence indicates that several species from the *Bacillus* genus, a group of spore-forming bacteria that have been utilized as probiotics, could act as robust mycotoxin biodegraders thanks to their high growth rate, resilience in harsh environmental conditions, and diverse metabolic pathways [26]. Therefore, prudent investigations of *Bacillus* species for their mycotoxin metabolisms are essential and may lead to the development of new mycotoxin controls, potentially enhancing food safety and security.

Although extensive research exists on the role of microorganisms in mitigating fungal toxins, there is a lack of comprehensive reviews or summaries specifically addressing their control by *Bacillus* bacteria. This manuscript therefore intends to evaluate studies from the past decade on the use of these species to degrade mycotoxins. It includes an overview of prevalent mycotoxins, their current detection methods and mitigation strategies, and notable studies on *Bacillus* species in metabolizing mycotoxins. This review concludes with perspectives on future directions and applications of the *Bacillus* bacteria, highlighting their promising roles as mycotoxin biodegraders.

## 2. Materials and Methods

This study aimed to document *Bacillus* species recognized for their ability to biodegrade mycotoxins. A comprehensive search of research and review articles was conducted using PubMed, Google Scholar, and Scopus electronic databases to obtain relevant original scientific studies. Key search terms included “*Bacillus*”, “degradation”, “detoxification”, and specific mycotoxins, such as “aflatoxin”, “zearalenone”, “T-2 toxin”, “fumonisin”, “deoxynivalenol”, “patulin”, and “ergot alkaloid”. These terms were searched individually and in combination, with the scope limited to the last ten years. Only peer-reviewed journal articles and the reputable scientific literature were considered. Abstracts were initially screened to assess the overall context of each publication. Only full-text manuscripts that met the aim of the study were included in this review. Duplicate and irrelevant studies were then removed, and each publication was thoroughly analyzed to ensure relevance and quality for the review of *Bacillus*-mediated degradation of fungal toxins.

### 2.1. Inclusion Factors

The following inclusion criteria were considered for this study:Recent publications between 2014 and 2024 were selected for this review. When further information was required, additional papers published before 2014 were also sought.Manuscripts that provided experimental evidence of biodegradation, such as toxin removal percentages, were included.Papers that discussed the degradation mechanisms, products of toxin metabolism, and toxicity studies were of great interest.

### 2.2. Exclusion Factors

The following criteria were applied to exclude certain studies:Research focusing solely on mycotoxin removal through cell wall binding in *Bacillus* species without involving actual biodegradation processes was excluded.Studies that utilized *Bacillus* bacteria to express toxin-degrading enzymes from other organisms were not considered.

## 3. Overview of Mycotoxins

There are approximately 150,000 fungal species formally described to date, and more than 500 mycotoxins have been reported in the literature [12,14]; these toxins are predominantly produced by fungi from the genera *Aspergillus*, *Fusarium*, *Byssochlamys*, *Penicillium*, *Claviceps*, and *Alternaria* [27,28]. Notably, certain fungal strains can biosynthesize multiple types of mycotoxins, and a single mycotoxin can be produced by multiple fungi [13]. The formation of fungal toxins depends on different factors, including temperature, humidity, pH, nutrients, inoculation level, physiological state, and microbial interactions [5]. Chemically, mycotoxins are mostly low-molecular-weight compounds with diverse structural classes, such as coumarins, alkaloids, terpenoids, and polyketides [29].

Mycotoxins are associated with a wide range of toxicities, such as carcinogenic, mutagenic, teratogenic, cytotoxic, neurotoxic, nephrotoxic, immunosuppressive, and estrogenic properties [11,29,30]. At the cellular level, certain mycotoxins can disrupt DNA and RNA synthesis through their interactions with nucleic acids [31]. The severity of these impacts is influenced by various factors, including the amount consumed, the length of exposure, and the possibility of synergistic interactions that could occur when different mycotoxins are ingested simultaneously [11].

While numerous fungal toxins exist, only a few are considered real threats to food safety [19]. The most agriculturally important and those of greatest concern are aflatoxins, ochratoxins, fumonisins, trichothecenes, zearalenone, and other classes, such as ergot alkaloids and patulin (Figure 1) [11]. Their fungal sources, toxicity, and occurrence are summarized in Table 1.

### 3.1. Aflatoxins (AFTs)

AFTs belong to a class of difuranocoumarins characterized by a bifuran group attached to the coumarin nucleus and either a pentanone ring or a lactone ring [32]. AFTs comprise approximately 20 structurally closely related compounds produced by various *Aspergillus* species, such as *A. flavus*, *A. parasiticus*, and, rarely, *A. nomius* [29]. Among the most significant AFTs are those belonging to series B and G, specifically aflatoxins B_1_ (AFB_1_) and B_2_ (AFB_2_), which are mainly produced by *A. flavus* and aflatoxins G_1_ and G_2_, commonly found in *A. parasiticus* (Figure 1) [33]. AFB_1_ is considered the most common food contaminant and also the most toxic aflatoxin [34]. Another notable member is aflatoxin M_1_, an AFB_1_ derivative excreted in the milk of cows and other ruminants that have consumed AFB_1_-contaminated feed [11]. Due to a strong link between their consumption and cancer occurrence, AFTs are officially recognized as human carcinogens, and their levels are strictly regulated in many countries [35]. AFTs mainly affect the liver, with several studies linking liver cancer to the presence of AFTs in food [15,36,37]. It is believed that the mechanisms of aflatoxin-induced carcinogenesis involve tumor promotion and progression, along with inducing various chromosomal aberrations, uncontrolled DNA synthesis, and chromosomal strand breaks in human cells [36]. AFTs are primarily encountered in cereals, rice, and corn, although they can also be detected in soybeans, sorghum, pistachio nuts, dried fruit, beer, milk, and spices [38].

### 3.2. Zearalenone (ZEN)

ZEN, also known as F-2 toxin (Figure 1), is a phenolic resorcylic acid lactone biosynthesized in a variety of soil fungi of the *Fusarium* family, including *F. graminearum*, *F. culmorum*, *F. cerealis*, *F. equiseti*, *F. crookwellense*, and *F. semitectum* [29]. ZEN is primarily found in mold-infested food and crops, with a predominant presence in grains, such as corn, wheat, rice, barley, sorghum, soybeans, oats, and related products [39]. Contamination of animal milk can occur when their feed contains elevated levels of ZEN [40]. ZEN is rapidly absorbed through the gastrointestinal tract after oral exposure and exhibits a wide range of harmful bioactivities, such as hepatotoxicity [41,42], immunotoxicity [43,44], and carcinogenicity [45,46]. Nonetheless, the most prominent and familiar effect of ZEN is its estrogenic activity. Along with its analogues, ZEN exerts a significant influence on estrogen receptors, possibly due to its structure resembling that of endogenous estrogen [29]. It has been shown that ZEN and its derivatives impact the synthesis and secretion of several steroid sex hormones, including testosterone, estradiol, and progesterone, through a series of mechanisms, such as affecting the activities of cellular mitochondria and steroidogenic enzymes [47]. ZEN significantly impacts the reproductive systems of livestock and animals [48]. These effects include reduced fertility, premature puberty, changes in thyroid, adrenal, and pituitary gland morphology, and fluctuations in serum progesterone and estradiol levels [49].

### 3.3. Ochratoxins (OTs)

OTs are polyketides derived from dihyrdocoumarins produced by several fungal species, including *A. ochraceus*, *A. carbonarius*, *A. niger*, *P. verrucosum*, *P. nordicum*, and *P. viridicatum* [29,50]. Ochratoxin A (OTA) is the most common and toxic among ochratoxins, which also include ochratoxin B (OTB) and ochratoxin C (OTC) (Figure 1) [29]. This toxin mainly targets the kidney and is suspected as a potential causal factor in certain kidney diseases, urothelial tumors, and chronic interstitial nephropathy [51]. Additionally, OTA is associated with mutagenic, teratogenic, neurotoxic, hepatotoxic, and immunotoxic effects [50,52]. It is commonly found in wheat, rye, coffee beans, nuts, raisins, wine, and pork and its derived products. Due to its prevalence in various commodities and its slow elimination from the body, OTA has been detected in the blood, breast milk, and urine, both in its original and metabolized states [11,51].

### 3.4. Fumonisins (FUMs)

FUMs feature an elongated hydrocarbon chain linked to several methyl and hydroxyl groups, along with one free amino and two tricarballylate functionalities, which are mainly responsible for their toxicity [53,54]. FUMs are predominantly produced by *Fusarium* fungal species, such as *F. verticillioides* and *F. proliferatum* [53]; however, *A. niger* can also synthesize certain FUMs [55]. There are at least 28 identified FUMs categorized into four groups (A, B, C, and P) based on structural similarities [11,53]. Regarding food safety, B series fumonisins (FBs) are of significance due to their frequent detection. Within this group, fumonisin B1 (FB_1_) and fumonisin B2 (FB_2_) (Figure 1) are especially noteworthy [53,56]. FBs induce a wide range of adverse effects in organisms, such as autophagy, apoptosis, neurotoxicity, immunotoxicity, reproductive toxicity, and carcinogenicity [57]. The primary mechanism through which FBs manifest their toxicity is by inhibiting ceramide synthase, thereby disrupting sphingolipid biosynthesis and impacting cellular growth, differentiation, and morphology [58].

### 3.5. Trichothecenes

Trichothecenes represent a diverse group of over 200 chemically related compounds produced by different filamentous fungal species, such as *Fusarium*, *Myrothecium*, *Trichoderma*, and *Trichothecium* [59,60]. They share a core structure of 12,13-epoxytrichothec-9-ene and are categorized into four types, A, B, C, and D, based on the substitution of the core skeleton [61]. Of primary concern for food safety are type A and B trichothecenes, predominantly produced by *Fusarium* species. Type A trichothecenes include T-2 toxin, which is regarded as the most studied and most toxic among all trichothecenes [59,62], together with HT-2 toxin and diacetoxyscirpenol (DAS) (Figure 1). Type B trichothecenes encompass deoxynivalenol (DON) and nivalenol (NIV) (Figure 1) [11,61]. DON, also known as vomitoxin, is also a well-studied mycotoxin due to its widespread occurrence in foodstuffs like cereal-based goods [63]; however, it is considered one of the least toxic trichothecenes. The presence of DON is often regarded as an indicator of other, more toxic trichothecenes and mycotoxins [63]. A notable characteristic of DON is its thermal stability, allowing it to withstand high temperatures for an extended period [64].

Trichothecenes are primarily found in cereals, such as wheat, rye, oats, and corn [65]. Their high level of cytotoxicity is expressed by their capacity to cause oxidative cell damage, inhibit nucleic acid and protein synthesis, interfere with cell division and mitochondrial operations, and compromise the stability of cell membranes [59,66]. The characteristic epoxide ring in their chemical structure is believed to be a primary contributor to their toxic effects [59]. Like other mycotoxins, trichothecenes cause a cascade of health impacts, such as hepatotoxicity, nephrotoxicity, neurotoxicity, immunotoxicity, and reproductive toxicity [67,68,69,70].

### 3.6. Other Prevalent Mycotoxins

Ergot alkaloids (EAs) are a group of mycotoxins derived from fungi of the genus *Claviceps*, most notably *C. purpurea* [71]. They can also be biosynthesized by certain *Epichloë* fungal species, such as *E. coenophiala* [72]. EAs belong to the class of indole alkaloids and are predominantly encountered in wheat, rye, and various cereals [73]. Over 80 EAs have been identified from a range of natural sources, primarily from *Claviceps* fungi, as well as from other fungal species and plants [71,74]. Notable EAs include ergometrine, ergocristine, ergosine, ergotamine, ergocornine, and ergocryptine, which are frequently detected [5,75]. EAs are known to interfere with the central nervous system [76], and exposure to these substances can result in serious psychological and physiological effects. In animals, EAs may lead to a decrease in productivity and severe health issues, including diarrhea, gangrene of the extremities, abortion, and internal bleeding [75,77].

Patulin (PAT) is another common mycotoxin produced by numerous members of the fungal genera *Penicillium*, *Aspergillus*, and *Byssochlamys*, predominantly by *P. expansum* [78,79]. Chemically, PAT is a small polyketide with a cyclic γ-lactone (Figure 1); it is water-soluble and has a low molecular weight [80]. PAT is commonly found in fruits and vegetables, especially in apples and apple products, as well as in other fruits, such as pears, cherries, grapes, and their products [78,81]. PAT can induce genotoxicity, immunotoxicity, and neurotoxicity in animals [82,83]. The toxicity is believed to result from interference of PAT with thiol-containing cellular components and amino acids in the plasma membrane [82,84], eventually leading to inhibition of protein and DNA syntheses and disruption of transcription and translation [81].

**Table 1 toxins-16-00478-t001:** Summary of key information regarding important mycotoxins.

Toxin	Fungal Source	Toxicity	Occurrence	Reference
AFTs	*A. flavus*, *A. parasiticus*, *A. nominus*, *A. niger*	Cancerogenic, teratogenic, mutagenic	Cereals, rice, corn, beer, milk, spices	[38]
ZEN	*F. graminearum*, *F. culmorum*, *F. cerealis*, *F. Equiseti*	Cytogenetic, embryotoxic, immunotoxic, estrogenic, antiandrogenic	Wheat, corn, rice, barley, sorghum, soybeans	[39]
DON	*F. graminearum*, *F. culmorum*	Immunotoxic, hematotoxic, neurotoxic	Grains, barley, wheat, corn	[63]
T-2	*F. sporotrichioides*, *F. poae*, *F. acuminatum*	Neurotoxic, hepatotoxic, immunotoxic, dermatotoxic	Wheat, barley, rye, oats, and maize	[59]
FUMs	*F. verticilloides*, *F. proliferatum*, *F. nygamai*	Neurotoxic, hepatotoxic, nephrotoxic	Corn, grains, wheat	[56]
EAs	*C. purpurea*, *C. fusiformis*, *E. coenophiala*	Neurotoxic, cardiotoxic	Wheat, rye, oats, barley	[85]
PAT	*P. expansum*, *A. clavatus*, *B. nivea*	Genotoxic, teratogenic, immunotoxic	Apples, pears, cherries, peaches, and their products	[86]

## 4. Current Detection and Decontamination Strategies of Mycotoxins

Because the permissible levels of mycotoxins in food are heavily regulated in many jurisdictions, the detection and quantification of mycotoxins require increasingly accurate analytical methods with progressively lower limits of detection (LODs) and limits of quantification (LOQs) [87,88]. A substantial body of literature, including various reviews, covers the detection methods for fungal toxins, ranging from fundamental analytical approaches to advanced detection techniques [7,89,90,91,92,93]. In essence, numerous analytical procedures have been developed and improved for the extraction, analysis, and quantification of mycotoxins. Chromatographic techniques, particularly thin-layer chromatography (TLC), high-performance liquid chromatography (HPLC), and gas chromatography (GC) integrated with mass spectrometry (MS), are frequently encountered. Spectroscopic methods, such as Fourier-transform infrared (FT-IR), and nuclear magnetic resonance (NMR) spectroscopy, have also been reported. Immunoassays, such as enzyme-linked immunosorbent assay (ELISA) and lateral flow immunoassay (LFIA), have been documented [94]. More recently, with the development of nanotechnology, biosensors and nanosensors have emerged as promising technologies for analyzing mycotoxins [95,96]. Figure 2A summarizes some popular techniques for detecting mycotoxins and their properties.

Regarding mycotoxin decontamination, various mitigation methods have been investigated aiming to minimize or eliminate their toxic effects on consumers’ health [97]. The current approaches to decontaminate mycotoxins include both conventional and innovative methods; they are generally classified into two main strategies, pre-harvest and post-harvest. Pre-harvest techniques focus on preventing the growth of toxigenic fungi in the fields to inhibit mycotoxin production [98]. Conversely, when mycotoxins have already been produced or their formation is unavoidable, post-harvest strategies are employed to remove or reduce their presence in food products (Figure 2B) [5,98].

### 4.1. Pre-Harvest

Pre-harvest strategies involve measures to prevent mold and fungal proliferation in crops and food products, coupled with ongoing monitoring for mycotoxins in agricultural produce and general products [99]. This can be achieved by implementing good agricultural practices (GAPs), good manufacturing practices (GMPs), as well as optimal storage conditions [98]. These strategies include executing crop selection, crop rotation schedules, and timing of cultivation and harvesting [100]. Applications of insecticides, fungicides, and herbicides to manage insect infestations, fungal infections, and weeds, as well as improving genetic traits to reduce mycotoxin production, are also widely used [99,101].

### 4.2. Post-Harvest

While preventing mycotoxin contamination in the field is recommended and represents an ideal strategy for mycotoxin control, it is important to note that pre-harvest methods do not ensure complete elimination of mycotoxins in food [102]. This is due to the extensive and persistent nature of fungal growth, which can occur despite the application of effective agricultural management and processing practices [8]. Thus, post-harvest strategies aim at reducing the fungal contamination and mycotoxin contents of agricultural products during storage, handling, processing, and transport [99]. Post-harvest controls of mycotoxins primarily comprise physical, chemical, and biological treatments.

#### 4.2.1. Physical Treatments

Physical techniques for mycotoxin decontamination mainly include sorting and separation, washing, solvent extraction, heating, irradiation, and adsorption of mycotoxins [103,104]. Sorting and separation are basic and simple methods that isolate toxin-contaminated foodstuffs from uncontaminated ones [105]. Washing and solvent extraction take advantage of the solubility of mycotoxins by using water or organic solvents to decontaminate them [106]. Heating involves thermal treatments to destroy heat-unstable mycotoxins; however, a number of fungal toxins, such as ZEN and DON, are thermally stable [107], making this method ineffective for their complete removal. Irradiation, which includes ionizing (X-ray and gamma-ray) and non-ionizing (ultraviolet and microwaves) techniques, has also been used for controlling mycotoxin contamination [108,109]. These radiations carry energy and thus can induce physical, chemical, and biological effects, thereby reducing or eliminating fungal growth and mycotoxins [5,106].

Adsorption, which utilizes binders that form complexes with mycotoxins, has been extensively explored for decades as a method to prevent mycotoxins from being absorbed by the gastrointestinal system [5,110]. Mycotoxin binders are classified as inorganic substances, like clays and activated carbon, and organic matrices, such as yeast cell walls [110]. The efficacy of binders in attaching to mycotoxins is influenced by several factors, including the molecular structures of the toxins, chemical interactions, such as hydrogen bonding, and physical attributes of the binders, like pore sizes [110]. Although these physical techniques have proven effective against a number of important mycotoxins, such as AFTs, trichothecenes, ZEN, and FBs [35,103,111,112,113], they usually carry drawbacks, such as high operational costs, potential reductions in the nutritional quality of food, and challenges in scaling up for large-scale applications [106].

#### 4.2.2. Chemical Treatments

Chemical techniques aim at altering or destroying the structures of mycotoxins, which generate less toxic or non-toxic products compared to the parent toxins. Decontamination of mycotoxins through chemical techniques generally involves alkaline, acid, oxidizing, and reducing agents [106,114]. For instance, ammonia is effective against AFB_1_, in which the lactone ring can be opened through base hydrolysis [115]. In the study by Aiko and coauthors, AFB_1_ was decomposed using lactic acid to AFB_2_ and AFB_2a_ which showed reduced cytotoxicity [116]. Ascorbic acid, or vitamin C, is another well-known mycotoxin-degrading chemical [117,118]. Doyle and coauthors found that patulin in apple juice fortified with 5% vitamin C was markedly degraded by 80% within 12 days [117]. Reducing agents, such as sodium bisulfite, have been used to destroy aflatoxins in corn [119]; its mechanism of action is believed to involve the formation of sulphonate derivatives that were later destroyed by heat [114]. Oxidizing agents, including ozone, hydrogen peroxide, sodium hypochlorite, and chlorine, have been used for mycotoxin removal [106] and have been proven effective against AFTs, DON, and ZEN [120,121,122]. However, similarly to physical controls, there are also growing concerns about chemical decontamination of fungal toxins, especially regarding the introduction or production of harmful by-products that are detrimental to animal and human health, as well as the environment [106].

#### 4.2.3. Biological Controls

Despite the development of various physical and chemical strategies to diminish or eradicate mycotoxins in agricultural commodities, the practical application of these techniques has been limited by their adverse effects [106]. Consequently, biodegradation of mycotoxins has emerged as a promising approach, attracting considerable research interest in recent years [123,124]. Biological methods involve the application of microorganisms or their metabolites for mycotoxin degradation and detoxification, offering an alternative approach to the control of mycotoxins [123]; this strategy is favored for its eco-friendliness, low cost, broad spectrum of target mycotoxins, and the potential of resulting in minimal or no toxic by-products or intermediates [5,123].

Numerous microorganisms, such as bacteria, yeasts, and fungi, have been identified for their ability to metabolize and remove mycotoxins [125]. Some of these organisms can degrade mycotoxins through enzymatic action by utilizing them as a carbon source [123]. Some act as organic binders or function both as adsorbers and as biodegraders [126]. Several fungi of the genus *Aspergillus* spp. were reported to have the ability to degrade and convert AFTs [127,128]. Likewise, FB_1_ and FB_2_ could be degraded by fungal species of *Saccharomyces* [129]. The yeast *Trichosporon mycotoxinovorans* was found to significantly degrade ZEN into non-toxic by-products [130]. Some bacterial strains of *Rhodococcus erythropolis*, *Flavobacterium aurantiacum*, *Pseudomonas putida*, *Escherichia coli*, and *Stenotrophomonas* sp. have also been found to effectively degrade AFB_1_ [131,132,133,134,135].

In recent years, many research efforts have concentrated on discovering microbial and recombinant enzymes that can break down fungal toxins. This is also a promising approach because certain microorganisms utilized for mycotoxin mitigation might produce harmful metabolites or may not tolerate the extreme conditions within the gastrointestinal tract of animals [106]. Enzymes, in contrast, are more specific and can generate harmless products that can potentially lead to total detoxification [125]. A wide range of macromolecules, such as oxidase, reductase, laccase, peroxidase, and carboxylesterases, along with other toxin-specific degrading enzymes, have been identified [136,137,138,139,140]. For instance, a recombinant ZEN-specific lactonohydrolase expressed in a *P. canescens* strain showed complete removal of ZEN in infected grain [141]. Cao and coauthors purified a new oxidase from the fungus *Armillariella tabescens*, which reacted with the bifuran ring of AFB_1_, reducing its toxicity [140]. Notably, among microorganisms capable of synthesizing active enzymes, species of the *Bacillus* genus are distinguished for their exceptional enzyme production efficacy, positioning them as strong candidates for mycotoxin biodegradation.

## 5. *Bacillus* Bacteria and Their Mycotoxin Control Potential

*Bacillus* is a large genus of Gram-positive, rod-shaped, spore-forming, aerobic or facultatively anaerobic bacteria [142] consisting of more than 300 species known to date [143,144]. These bacteria are ubiquitously distributed across various environments and are commonly encountered in soils; however, they also inhabit water, air, plant surfaces, and the gastrointestinal tracts of humans and animals [26,142]. Due to the diversity in genomic and phenotypic traits among its species, this genus has undergone several taxonomic revisions, resulting in its current classification into distinct clades [143]. One prominent clade is the *B. cereus* group, which consists of closely related *Bacillus* species, such as *B. cereus*, *B. anthracis*, and *B. cytotoxicus*; members of this group are often considered pathogenic and have been implicated in certain diseases, including anthrax and foodborne illnesses [145]. Conversely, the *B. subtilis* group comprises species like *B. subtilis*, *B. licheniformis,* and *B. pumilus*, which are recognized for their non-pathogenic nature and potential for biotechnological applications [143].

*Bacillus* species are renowned for their remarkable ability to withstand extreme environmental conditions, such as acids and bile salts in the gastrointestinal tracts, heat processing, and low-temperature storage [146,147]. This resilience is largely due to their capacity to form endospores, a complex developmental process where the bacterial cells differentiate into resistant spores, allowing their survival in extreme temperatures, variations in pH, ultraviolet radiations, harmful chemicals, and nutrient shortage [148]. Upon the restoration of favorable conditions, the spores germinate back into vegetative cells capable of growth and reproduction [148,149]. Additionally, this genus is proficient in bio-synthesizing a wide array of proteins, enzymes, and small molecules with diverse structures, such as cyclic lipopeptides, surfactins, iturins, and fengycins, which exhibit notable bioactivities, including antibacterial, antifungal, and anticancer activities [144,150,151,152]. Consequently, *Bacillus* species are valuable for extensive application across multiple areas, such as the food industry and medicine [149].

The applications of these bacteria in mitigating fungal toxins have been extensively studied, with hundreds of publications, reporting both *in vitro* and *in vivo* studies, available in scientific databases. Research on mycotoxin control using *Bacillus* bacteria generally falls into three main categories. The first one utilizes the bacteria to inhibit the growth of mycotoxigenic fungi, thus preventing the in situ biosynthesis and accumulation of mycotoxins. For instance, it was discovered that the secondary metabolites from *B. subtilis* BS-Z15 caused downregulation in genes responsible for cellular reproduction and aflatoxin synthesis in *A. flavus*, thus preventing fungal growth and significantly reducing aflatoxin production [153]. Similarly, a detailed literature analysis by Veras and coauthors indicated that numerous *Bacillus* strains exerted antifungal activity against toxin-producing fungi [26].

The second category utilizes the *Bacillus* strains to degrade or metabolize mycotoxins, thereby reducing their concentrations. This research direction has received considerable attention recently. Indeed, search queries on electronic databases returned more than 150 research articles published between 2014 and 2024 regarding this topic, nearly triple the number of papers from the previous decade. The articles were reviewed, and their information was extracted and is presented in Figure 3, which illustrates the capacity of *Bacillus* strains to degrade different types of mycotoxins.

As shown in Figure 3, *B. subtilis* were the most extensively studied, with over 80 different strains capable of metabolizing various types of mycotoxins, followed by *B. velezensis*, *B. amyloliquefaciens*, and *B. licheniformis*. In addition, AFTs and ZEN were the mainly targeted toxins. It also appears that almost every important mycotoxin is known to be degradable by at least one *Bacillus* strain. For example, AFB_1_ was degraded by *B. velezensis* DY3108, *B. shackletonii* L7, and *B. licheniformis* CFR1, with degradation rates of 91.5, 92.1, and 94.7%, respectively [154,155,156]. *B. licheniformis* YB9 and *B. subtilis* ASAG 216 were found to remove up to 83 and 81% of DON [157,158]. Liu and coauthors reported the biodegradation of ZEN by *B. spizizenii* B73 by 99% [159].

While most research has reported the general capability of mycotoxin biodegradation by *Bacillus* spp., only a few studies have conducted more in-depth investigations, such as identifying specific metabolic pathways, detecting products of metabolism, and assessing the toxicity of the end products. Table 2 provides an overview of these studies, highlighting the key findings, and Figure 4 shows the chemical structures of known and proposed metabolized products of mycotoxins by *Bacillus* bacteria. It can be seen that enzymatic degradation represented the primary mechanisms through which *Bacillus* bacteria metabolized mycotoxins, with the discovery of several types of enzymes, such as esterase, laccase, phosphotransferase, carboxypeptidase, and peroxidase, responsible for the degradation [159,160,161,162,163,164,165,166,167,168,169,170]. These enzymes effectively degraded, destroyed, or transformed the toxin structures into end products through multiple different actions, such as cleaving the ester bonds, opening the epoxide rings, or attaching functional groups to the core skeletons. For instance, Guo and coauthors identified a laccase from *B. licheniformis* ANSB821 that could oxidize AFB_1_ to AFQ_1_ (compound **4** in Figure 4). Similarly, Yang and coauthor observed that ZEN was completely transformed into ZEN-14-phosphate (compound **31** in Figure 4) through an intracellular phosphotransferase produced by *B. subtilis* Y816 [161]. Toxin degradation by *Bacillus* bacteria, in some instances, has also been observed to include both enzymatic pathways and physical adsorption [171]. For example, in Zhang’s study, OTA was removed both by enzymes in the cell-free supernatant and by binding to the peptidoglycan layers of the bacterial cell wall of *B. velezensis* E2 [172].

Toxicity assessments of the biodegraded products have become more common in recent publications, often utilizing various evaluation methods. Among these, cytotoxicity testing of the products against diverse cell lines is one of the most frequently employed methods (Table 2) [160,173,174]. Generally, most studies reported a reduction in toxicity of the end products. However, despite this reduction, some residual toxicity often remains, and many other degradation products have unknown toxicity profiles. In rare cases, degradation of mycotoxins by *Bacillus* spp. has resulted in detoxification by producing known non-toxic compounds. For example, Li and coauthors found that *Bacillus* sp. LS100 could degrade DON to deepoxy-deoxynivalenol, DOM-1 (compound **20** in Figure 4) [175]. DOM-1 is well-regarded as a much less toxic metabolite of DON due to the lack of the epoxide ring responsible for the toxicity [176]. A study on porcine intestinal epithelial cells revealed that DOM-1 showed no cytotoxicity, while DON caused more than 40% cell death at the same concentration [177].

**Table 2 toxins-16-00478-t002:** Representative studies of biodegradation of important mycotoxins by *Bacillus* species.

Toxin	*Bacillus* Strains	Toxin Metabolism Pathways	Degradation Efficiency	Metabolized Products (Figure 4)	Toxicity Study of Metabolized Products	Ref.
AFB_1_	*B. subtilis* B-59994	Enzymes in cell-free extracts	60% in 100 h	1–3	Not mentioned	[178]
	*B. subtilis* SCK6	BsDyP peroxidase	77% in 48 h	2	Not confirmed	[179]
	*B. licheniformis* ZOM-1	CotA laccase	90% in 24 h	2, 4	GES-1 viability, 100%, as against 75% for AFB_1_ at 4 ppm	[160]
	*B. licheniformis* ANSB821	CotA laccase	96% in 12 h	4	L-02 viability, 100%, as against 40% for AFB_1_ at 100 µM	[180]
	*B. subtilis*	CotA laccase	100% in 12 h			[181]
	*B. subtilis*	CotA laccase	80% in 72 h			[182]
	*Bacillus* sp. H16v8 and *Bacillus* sp. HGD9229	Enzymes in cell-free supernatants of culture	57% in 12 h	5, 6	Not confirmed	[183]
	*B. subtilis* UTB1	BacC oxidoreductase	69% in 7 d	7, 8	Not mentioned	[184]
	*B. megaterium* HNGD-A6	Lactonase	95% in 72 h	8	Hep-G2 viability, 85%, as against 62% for AFB_1_ at 30 µg/mL	[185]
	*B. amyloliquefaciens* ZG08	Thiol peroxidase and glycerol dehydrogenase	81% in 72 h			[186]
	*B. albus* YUN5	Enzymes in cell-free supernatants of culture	55% in 120 h	8–10	HCC viability, 69%, as against 57% for AFB_1_	[187]
	*B. aryabhattai*	Enzymes in cell-free supernatants of culture	83% in 72 h	11	L-02 viability, 80%, as against 50% for AFB_1_ at 7.4 µg/mL	[188]
	*B. amyloliquefaciens* YUAD7	Enzymes in cell-free supernatants of culture	92% in 72 h	12–15	L-02 viability, 92%, as against 40% for AFB_1_ at 10 µg/mL	[189]
AFM_1_	*B. pumilus* E-1-1-1	Catalase	63% in 12 h	16	Hep-G2 viability, 50%, as against 20% for AFM_1_ at 200 ng/mL	[173]
DON	*Bacillus* sp. HN117	Not mentioned	29% in 72 h	17	Not mentioned	[190]
	*Bacillus* sp. N22	Not mentioned	21% in 120 h	18, 19	Not mentioned	[190]
	*Bacillus* sp. LS100	Not mentioned	100% in 72 h	20	IPEC-J2 viability, 100%, as against 60% for DON at 7 µM	[175,177]
	*B. subtilis* SCK6	BsDyP peroxidase	78% in 48 h	21	Not confirmed	[179]
OTA	*B. subtilis* ANSB168	Carboxypeptidases	47% in 18 h	22	Zebrafish embryo lethality, 10%, as against 100% for OTA at 0.5 µM 3 days post-fertilization	[191,192]
	*B. velezensis* IS-6	Hydrolase	89% in 24 h			[193]
	*B. subtilis*	Metalloendopeptidase	45% in 1 h			[194]
	*B. velezensis* E2	Enzymatic degradation and alkaline hydrolysis	96% in 48 h	23	Not mentioned	[172]
ZEN	*B. spizizenii* B73	Lactonehydrolase	99% in 8 h	24	Not mentioned	[159]
	*B. pumilus* ES-21	Esterase	99% in 48 h			[195]
	*B. subtilis YT-4*	Lactonehydrolase	95% in 36 h	25, 26	Not mentioned	[196]
	*B. amyloliquefaciens* D-1	Ester hydrolyzation and amino acid esterification	96% in 24 h	27	Hep-G2 viability, 100%, as against 50% for ZEN at 20 µM	[174]
	*B. subtilis* YQ-1	Glycosyltransferase	98% in 16 h	28, 29	MCF-7 proliferation, 40%, as against 60% for ZEN at 50 µg/mL	[197]
	*B. licheniformis* ZOM-1	CotA laccase	98% in 24 h	30	GES-1 viability, 100%, as against 62% for ZEN at 10 ppm	[160]
	*B. subtilis* SCK6	BsDyP peroxidase	85% in 48 h			[179]
	*B. subtilis* Y816	Phosphotransferase	100% in 12 h	31	Not confirmed	[161]
	*Bacillus* sp. S62-W	Not mentioned	100% in 24 h			[198]
T2	*Bacillus* sp.	Intracellular enzymes	99% in 12 h	32	IC_50_ 2.0 µM, as against 0.5 µM for T2 toxin in NHLF	[199,200]

GES-1: human gastric epithelial cell line; L-02: human hepatic cell line; Hep-G2: human liver cancer cell line; HCC: human hepatocellular carcinoma cell line; MCF-7: human breast cancer cell line; IPEC-J2: porcine intestinal epithelial cells; NHLF: normal human lung fibroblasts.

The third research category of *Bacillus* spp. in mitigating mycotoxins is countering the mycotoxicosis in vivo. This is illustrated mainly in animal studies, especially in livestock and poultry animals, where the use of *Bacillus* species has shown promising results in enhancing overall animal health, with most research reporting the bacteria alleviating the toxicities induced by mycotoxins, as well as additional benefits, such as regulating the gut microbiota. For instance, Zhang and coauthors investigated the potential of *B. subtilis* ZJ-2019-1 in attenuating the toxic effects of ZEN on female gilts [201], in which gilts with ZEN toxicosis resulted in significantly impaired reproductive organs and liver function, altered serum hormone levels, and histopathological damages in the liver, uterus, and ovaries; however, administration of feed containing *B. subtilis* ZJ-2019-1 countered the harmful impacts of the toxin, including, in particular, reducing the presence of ZEN and its metabolites in the serum and urine, alleviating organ damage, and improving hormone levels [201]. Other similar in vivo research on utilizing *Bacillus* bacteria to mitigate mycotoxicosis in farm animals is summarized in Table 3.

## 6. Conclusions and Perspectives

Mycotoxins continue to pose a severe threat to both human and animal health, as well as to the global economy; this therefore requires the development of effective control strategies to counter their impacts. Among the various approaches explored above, microbial degradation of mycotoxins utilizing bacteria of the *Bacillus* genus has emerged as a highly promising solution. These bacteria have demonstrated a robust capacity to metabolize fungal toxins with their diverse metabolic pathways, suggesting their potential as robust candidates for mycotoxin management across various areas.

However, a few aspects should be considered for further research on these bacteria. Firstly, most studies on *Bacillus* spp. focus on the biodegradation of a single mycotoxin, while in the real-world environment, different types of mycotoxins often coexist [221], and thus addressing only one mycotoxin may not achieve a comprehensive picture of toxin removal. Therefore, future research should make efforts at discovering strains capable of targeting multiple mycotoxins to warrant better decontamination.

Secondly, toxicity is the primary concern of fungal toxins, and complete detoxification is the ultimate objective of their control. Some studies mistakenly assume that degradation is detoxification, which can lead to misconceptions about the effectiveness of biodegradation in reducing toxicity [54]. It is essential to recognize that transformation or degradation of mycotoxins does not always result in detoxification, as degradation products can sometimes still be as toxic as the original toxin [222]. For example, biodegraded products of ZEN, such as α-, β-zearalenol, or ZEN-glucosides, are known to retain similar toxicity [223]. Moreover, some *Bacillus* strains, despite their ability to degrade toxins, can in turn produce harmful enterotoxins, creating further complications [224]. While many research findings on *Bacillus* species confirm their ability to transform fungal toxins, there is limited information regarding the toxicity of the resulting metabolized products, as well as the potential of these bacteria to produce enterotoxins. Consequently, research on *Bacillus* should prioritize not only the biotransformation but also the detoxification of mycotoxins and the identification of non-toxic strains.

The implementation of *Bacillus* species in the agricultural sector, particularly as feed additives and probiotics, to detoxify mycotoxins could offer a promising measure to protect farm animals. This approach potentially not only enhances food safety but also promotes the overall health and productivity of livestock, ultimately contributing to a more sustainable and secure food supply chain. However, these applications are often hindered by the lack of mechanistic studies on toxin biotransformation, such as identifying the toxin-degrading enzymes and their properties and stability. Thus, further understanding of the mechanisms of action and validation of *Bacillus* and their products in controlling mycotoxins would hold significant promise for safeguarding both animal welfare and agricultural productivity.

## Figures and Tables

**Figure 1 toxins-16-00478-f001:**
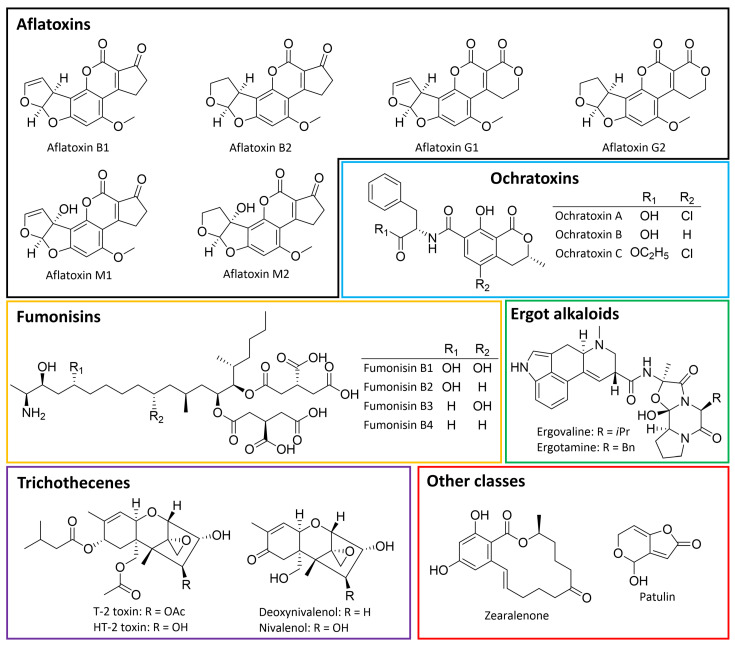
Chemical structures of important representative mycotoxins.

**Figure 2 toxins-16-00478-f002:**
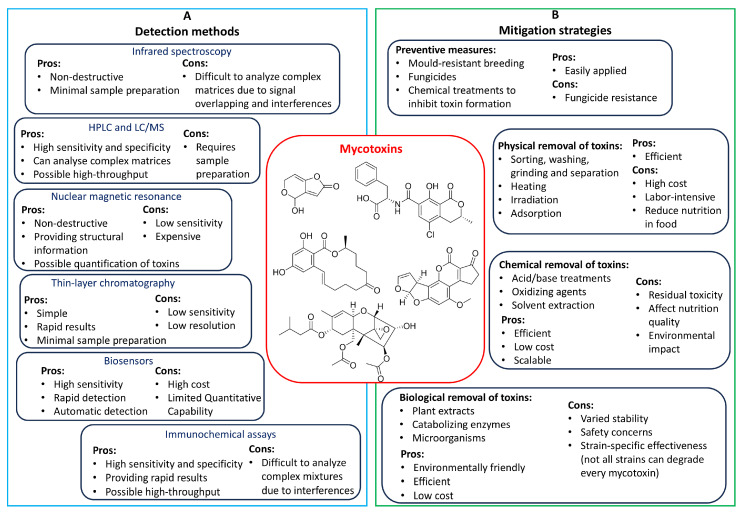
Commonly used methodologies in mycotoxin studies. (**A**). Common detection methods. (**B**). Mitigation strategies for mycotoxins.

**Figure 3 toxins-16-00478-f003:**
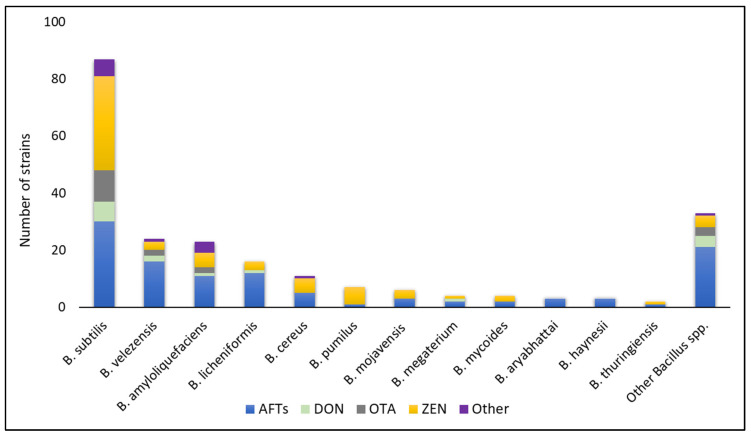
Summary of *Bacillus* spp. and the ability to degrade mycotoxins. Data were compiled from relevant publications on mycotoxin biodegradation by Bacillus bacteria from 2014 to 2024.

**Figure 4 toxins-16-00478-f004:**
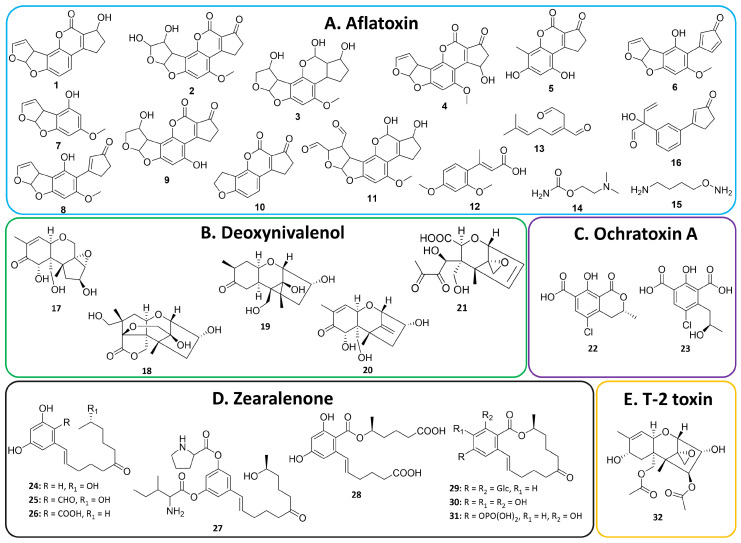
Confirmed and proposed chemical structures of biotransformation products of mycotoxins by *Bacillus* spp.

**Table 3 toxins-16-00478-t003:** In vivo studies on the mitigation of mycotoxicosis in farm animals by *Bacillus* species.

*Bacillus* Species	Toxins	Animals	Observed Outcomes After *Bacillus* spp. Supplement	References
*B. subtilis*	1	Broiler chicken	Decreasing toxicities on the kidney and the immune system.	[202]
*B. subtilis* ANSB01G	2	Gestating sows	Improving reproductive performance and alleviating toxicity of the toxin.	[203]
*B. subtilis* and *B. licheniformis*	1, 3	Broiler chicken	Improving feed consumption and weight gain during the growing stage.	[204]
*B. subtilis* and *B. licheniformis*	1, 4	Broiler chicken	Improving feed conversion ratios, reducing biochemical alterations, and decreasing *E. coli* counts in the caeca.	[205]
Compound probiotic containing *B. subtilis*	4	Broiler chicken	Improving animal growth and reducing residual toxin in serum, excreta, and liver.	[206]
*B. subtilis* ASAG 216	5	Piglets	Attenuating toxin-induced inflammation and oxidative stress.	[207]
*B. subtilis* ANSB168	1	Laying hens	Alleviating toxin-induced inflammation.	[191]
*B. subtilis* ZJ-2019-1	2	Gilts	Alleviating organ damage and improving hormone levels.	[201]
*B. subtilis*, *B. megaterium*, and *B. laterosporus*	4	Quails	Increasing carcass yield and enhancing immune responses, antibody production, and bone health.	[208]
Compound probiotics containing *B. subtilis*	2, 4	Broiler chicken	Increasing production performance, stabilizing gut microbiota, and alleviating histological lesions.	[209]
*B. subtilis* ANSB01G	2	Gestating sows	Alleviating toxin-induced apoptosis, oxidative stress, and histological damages of organs; reducing toxin residues in feces.	[210]
*B. subtilis* SP1 and *B. subtilis* SP2	2	Pigs	Regulating gut microbiota and decreasing toxin concentration.	[211]
*B. subtilis* ANSB060	4	Dairy cow	Reducing toxin bioavailability; decreasing excretion of toxin products in milk.	[212]
*B. subtilis* ANSB060	4 and other AFTs	Broiler chicken	Reducing toxin level in the duodenum and liver, improving growth performance, and enhancing meat quality.	[213,214]
*B. subtilis* ANSB060	4	Laying hens	Improving eggshell strength and ameliorating liver and kidney damage.	[215]
*B. subtilis* ANSB060	AFTs	Ducks	Increasing average daily gain and antioxidant enzyme functions; reducing toxin level in liver.	[216]
*B. subtilis*	2 and AFTs	Laying hens	Improving egg production and feed intake.	[217]
*B. licheniformis* CK1	2	Piglets	Normalizing reproductive organ weight and hormone levels.	[218,219]
*B. subtilis* ANSB01G	2, 5	Gilts	Regulating immune function and reproductive health.	[220]
*Bacillus* sp. LS100	5	Swine	Improving daily feed consumption, weight gain, and feed efficiency.	[175]

**1**: OTA, **2**: ZEN, **3**: T2 toxin, **4**: AFB_1_, **5**: DON.

## Data Availability

The original contributions presented in this study are included in the article. Further inquiries can be directed to the corresponding author.

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
