# Peer review of "Mycotoxin Biodegradation by Bacillus Bacteria—A Review"

_toxins, 2024, doi:10.3390/toxins16110478_

Round 1
Reviewer 1 Report
Comments and Suggestions for Authors
In this review article, the author presents a study that addresses mycotoxins pose significant health risks and their economic challenges in the food industry. In recent years, the biodegradation of mycotoxins using microorganisms has attracted much attention due to its reliability, efficiency, and cost-effectiveness. This review article aims to provide information on Bacillus bacteria's biodegradation of fungal toxins, signifying their potential applications in developing mycotoxin-degrading products. Some corrections required in this paper are mentioned below.
1) Some corrections with their correct form and respective lines are mentioned below.
Foodstuff – foodstuffs line 24
Public - publics line 34
Human - humans line 35
Comprises- comprise line 100
Pituitary glands: pituitary gland line 132
Proteins-protein line 177
Decontaminate-decontaminates line 257
Cell wall-cell walls line268
Reduction -reductions line274
Groups- group line 334
Rate-rates line 375
2) In line 53, the statement section ‘focused exploring innovative biological’ is correct with ‘focused on exploring innovative biological’.
3) In line 284, the statement section ‘involve formation’ is correct with ‘involve the formation’.
4) In line 303, the word 'souce` is correct with 'source`.
5) In line 350, the statement section ‘as food industry’ is correct with ‘as the food industry’.
6) In line 367, ‘information extracted’ is correct with ‘information was extracted’.
7) In line 390, `both enzymatic pathway` is correct with `both the enzymatic pathway`.
8) In line 428, `biodegradation on a single` is correct with `biodegradation of a single'.
9) In line 429, `while in real world` is correct with `in a real world`.
10) In line 452,'mechanism studies` is correct with 'mechanism of studies`.
Comments on the Quality of English LanguageMinor language corrections required.
Author Response
Reviewer 1:
In this review article, the author presents a study that addresses mycotoxins pose significant health risks and their economic challenges in the food industry. In recent years, the biodegradation of mycotoxins using microorganisms has attracted much attention due to its reliability, efficiency, and cost-effectiveness. This review article aims to provide information on Bacillus bacteria's biodegradation of fungal toxins, signifying their potential applications in developing mycotoxin-degrading products. Some corrections required in this paper are mentioned below.
1) Some corrections with their correct form and respective lines are mentioned below.
Foodstuff – foodstuffs line 24
Public - publics line 34
Human - humans line 35
Comprises- comprise line 100
Pituitary glands: pituitary gland line 132
Proteins-protein line 177
Decontaminate-decontaminates line 257
Cell wall-cell walls line268
Reduction -reductions line274
Groups- group line 334
Rate-rates line 375
2) In line 53, the statement section ‘focused exploring innovative biological’ is correct with ‘focused on exploring innovative biological’.
3) In line 284, the statement section ‘involve formation’ is correct with ‘involve the formation’.
4) In line 303, the word 'souce` is correct with 'source`.
5) In line 350, the statement section ‘as food industry’ is correct with ‘as the food industry’.
6) In line 367, ‘information extracted’ is correct with ‘information was extracted’.
7) In line 390, `both enzymatic pathway` is correct with `both the enzymatic pathway`.
8) In line 428, `biodegradation on a single` is correct with `biodegradation of a single'.
9) In line 429, `while in real world` is correct with `in a real world`.
10) In line 452,'mechanism studies` is correct with 'mechanism of studies`.
Author reply:
We agree with the reviewer and have corrected the wording as suggested. The changes are highlighted in the revised manuscript.
We thank the reviewer for their time and suggestions for making the manuscript better.
Reviewer 2 Report
Comments and Suggestions for Authors
October 17th, 2024
Letter to Authors
Dear Authors,
I have read and reviewed your manuscript titled “Mycotoxin Biodegradation by Bacillus Bacteria - A Review” (toxins- 3220851) submitted to TOXINS.
Manuscript presents valuable review about mycotoxins (classes, method of detection and decontamination). The review is writting well and its idea is very important. The biodegradation of fungal toxins by Bacillus bacteria, known, among other things, for their probiotic properties, points to their potential use in the development of mycotoxin-degrading products. This work seemed to me very interesting and multifaceted.
After reading of your manuscript I could recommend it to be published after attending the REVISIONS. In order to improve the quality of the manuscript I suggest following corrections:
Comments:
Chapter 1. Introduction. No comments.
Chapter 2. Material and Methods. The methods section is missing, in particular the search methodology, exclusion factors, etc.
Chapter 3 and 4
1. In chapter “3.2.2. Chemical Treatments” does not mention such a well-known substance as ascorbic acid, as both an acidic and oxidising agent and used to remove mycotoxins, e.g. in fruit products, although in the manuscript there are references on ascorbic acid (80 and 81 in the literature list), but they are quoted quite elsewhere.
2. Although there are several examples in the body of the work, at the end of Chapter 3 it would be crucial and valuable to collate, in the form of a table or diagram, the effectiveness of methods (physical, chemical, biological) in removing mycotoxins, according to the classes listed in Figure 1 and with examples.
3. Minor editorial corrections: All Latin names should be written in italics (e.g. body of Fig. 3, caption of figure 3 and 4) – check the entire manuscript.
Conclusions and Perspectives. No comments.
References. The paper is very rich and has a great deal of literature, but much of it lacks DOI (e.g. 5,6,14,16,17,30,33,36,44,86,88,120,179,183,188,195,207,212,215, and more). Please carefully check and complete all DOI.
I really thank you for your consideration, and I sincerely hope these recommendations could be useful to you to improve the quality of your manuscript since the item might be important to potential readers of Toxins.

Author Response
Reviewer 2:
I have read and reviewed your manuscript titled “Mycotoxin Biodegradation by Bacillus Bacteria - A Review” (toxins- 3220851) submitted to TOXINS.
Manuscript presents valuable review about mycotoxins (classes, method of detection and decontamination). The review is writing well and its idea is very important. The biodegradation of fungal toxins by Bacillus bacteria, known, among other things, for their probiotic properties, points to their potential use in the development of mycotoxin-degrading products. This work seemed to me very interesting and multifaceted.
After reading of your manuscript I could recommend it to be published after attending the REVISIONS. In order to improve the quality of the manuscript I suggest following corrections:
Comments:
Chapter 1. Introduction. No comments.
Chapter 2. Material and Methods. The methods section is missing, in particular the search methodology, exclusion factors, etc.
Author reply:
We have included the material and methods section in the revised manuscript as follow:
“2. Materials and methods
This study aimed to document Bacillus species recognized for their ability to biodegrade mycotoxins. A comprehensive search of research and review articles was conducted, using PubMed, Google Scholar, and Scopus electronic databases to obtain relevant original scientific studies. Key search terms included: “Bacillus”, “degradation”, “detoxification”, and specific mycotoxins such as “aflatoxin”, “zearalenone”, “T-2 toxin”, “fumonisin”, “deoxynivalenol”, “patulin”, and “ergot alkaloid”. These terms were searched individually and in combination, with the scope limited to the last ten years. Only peer-reviewed journal articles and reputable scientific literature were considered. Abstracts were initially screened to assess the overall context of each publication. Only full-text manuscripts that met the aim of the study were included in this review. After removing irrelevant and duplicate studies, a total of 130 publications were selected for the review of Bacillus-mediated degradation of fungal toxins; each publication was thoroughly analyzed to ensure relevance and quality.
2.1 Inclusion factors
The following inclusion criteria were considered for this study on biodegradation of mycotoxins by Bacillus species:
- Recent publications between 2014 and 2024 were selected for this review.
- Manuscripts that provided experimental evidence of biodegradation, such as toxin removal percentages.
- Papers that discussed the degradation mechanisms, products of toxin metabolism, and toxicity studies were of great interest.
2.2 Exclusion factors
The following criteria were applied to exclude certain studies:
- Research focusing solely on mycotoxin removal through cell wall binding in Bacillus species, without involving actual biodegradation processes, was excluded.
- Studies that utilized Bacillus bacteria to express toxin-degrading enzymes from other organisms were not considered.”
Chapter 3 and 4
- In chapter “3.2.2. Chemical Treatments” does not mention such a well-known substance as ascorbic acid, as both an acidic and oxidising agent and used to remove mycotoxins, e.g. in fruit products, although in the manuscript there are references on ascorbic acid (80 and 81 in the literature list), but they are quoted quite elsewhere.
Author reply:
As suggested, we have elaborated ascorbic acid as a well-known chemical for mycotoxin removal in the revised manuscript. Specifically, we have added the following statement to the chemical treatment section:
“Ascorbic acid, or vitamin C, is another well-known mycotoxin-degrading chemical.[117, 118] Doyle and coauthors found that patulin in apple juice fortified with 5% vitamin C was markedly degraded by 80% within 12 days.[117]”
Two new references for the use of vitamin C as a chemical treatment for mycotoxins have also been included in the reference:
- Doyle, M.P.; Applebaum, R.S.; Brackett, R.E.; Marth, E.H. Physical, Chemical and Biological Degradation of Mycotoxins in Foods and Agricultural Commodities. Journal of Food Protection 1982, 45, 964-971, DOI: https://doi.org/10.4315/0362-028X-45.10.964.
- Brackett, R.E.; Marth, E.H. Ascorbic Acid and Ascorbate Cause Disappearance of Patulin from Buffer Solutions and Apple Juice. Journal of Food Protection 1979, 42, 864-866, DOI: https://doi.org/10.4315/0362-028X-42.11.864.
- Although there are several examples in the body of the work, at the end of Chapter 3 it would be crucial and valuable to collate, in the form of a table or diagram, the effectiveness of methods (physical, chemical, biological) in removing mycotoxins, according to the classes listed in Figure 1 and with examples.
Author reply:
We appreciate the reviewer’s suggestion to include a summary table or diagram on the degradation methods of mycotoxins, and we understand such a visualization could be useful for the readers. However, in the original manuscript, we have already provided a detailed discussion of the degradation methods in the body of the manuscript, with each method carefully explained. We also had a figure (Figure 2) discussing the mitigation methods for mycotoxins. Additionally, several review articles in the literature on the physical, chemical, and biological controls of mycotoxins already present comprehensive summary tables on the degradation methods. Constituently, we believe including a table might result in redundancy, which repeats the contents already outlined.
Nonetheless, instead of presenting a summary table, we have updated Figure 2 in the revision. Specifically, we have incorporated additional details outlining the advantages and disadvantages of each degradation method. We believe this update would provide equivalent information and achieve the same purpose as the suggested table or diagram.
- Minor editorial corrections: All Latin names should be written in italics (e.g. body of Fig. 3, caption of figure 3 and 4) – check the entire manuscript.
Author reply:
We have checked the entire manuscript and italicized the Latin names accordingly.
Conclusions and Perspectives. No comments.
References. The paper is very rich and has a great deal of literature, but much of it lacks DOI (e.g. 5,6,14,16,17,30,33,36,44,86,88,120,179,183,188,195,207,212,215, and more). Please carefully check and complete all DOI.
Author reply:
We have carefully checked all the references and included the DOI at the end of each reference. We have also updated the reference style to comply with MDPI referencing standards.
I really thank you for your consideration, and I sincerely hope these recommendations could be useful to you to improve the quality of your manuscript since the item might be important to potential readers of Toxins.
Author reply:
We appreciate the valuable feedback of the reviewer; we thank the reviewer for their time and efforts in reviewing the manuscript.
Round 2
Reviewer 2 Report
Comments and Suggestions for Authors
I accept the manuscript in present form.